# Contrastive Multiview Coding

## Abstract

Humans view the world through many sensory channels, e.g., the long-wavelength light channel, viewed by the left eye, or the high-frequency vibrations channel, heard by the right ear. Each view is noisy and incomplete, but important factors, such as physics, geometry, and semantics, tend to be shared between all views (e.g., a "dog" can be seen, heard, and felt). We hypothesize that a powerful representation is one that models view-invariant factors. Based on this hypothesis, we investigate a contrastive coding scheme, in which a representation is learned that aims to maximize mutual information between different views but is otherwise compact. Our approach scales to any number of views, and is view-agnostic. The resulting learned representations perform above the state of the art for downstream tasks such as object classification, compared to formulations based on predictive learning or single view reconstruction, and improve as more views are added. On the Imagenet linear readoff benchmark, we achieve $68.4\%$ top-1 accuracy.

## 1 Introduction

A foundational idea in coding theory is to learn compressed representations that nonetheless can be used to reconstruct the raw data. This idea shows up in contemporary representation learning in the form of autoencoders (Salakhutdinov & Hinton, 2009) and generative models (Kingma & Welling, 2013; Goodfellow et al., 2014), which try to represent a data point or distribution as losslessly as possible. Yet lossless representation might not be what we really want, and indeed it is trivial to achieve – the raw data itself is a lossless representation. What we might instead prefer is to keep the "good" information (signal) and throw away the rest (noise). How can we identify what information is signal and what is noise?

To an autoencoder, or a max likelihood generative model, a bit is a bit. No one bit is better than any other. Our conjecture in this paper is that some bits *are* in fact better than others. Some bits code important properties like semantics, physics, and geometry, while others code attributes that we might consider less important, like incidental lighting conditions or thermal noise in a camera's sensor.

We hypothesize that the good bits are the ones that are shared between multiple *views* of the world, for example between multiple sensory modalities like vision, sound, and touch. Under this perspective "presence of dog" is good information, since dogs can be seen, heard, and felt, but "camera pose" is bad information, since a camera's pose has little or no effect on the acoustic and tactile properties of the imaged scene. There is significant evidence in the cognitive science and neuroscience literature that such cross-view representations are encoded by the brain (e.g., Smith & Gasser (2005); Den Ouden et al. (2012); Hohwy (2013)).

Our goal is therefore to learn representations that capture information shared between multiple sensory views but that are otherwise compact (i.e. throw away the bad information). To do so, we employ contrastive learning, where we learn a feature embedding such that views of the same scene map to nearby points while views of different scenes map to far apart points. In particular, we adapt the recently proposed method of Contrastive Predictive Coding (CPC) (Oord et al., 2018), except we simplify it – removing the recurrent network – and generalize it – showing how to apply it to arbitrary collections of views, rather than just to temporal predictions. In reference to CPC, we term our method *Contrastive Multiview Coding* (CMC). The contrastive objective in our formulation, as in CPC, can be understood as attempting to maximize the mutual information between the representations of each view.

We intentionally leave "good bits" only loosely defined and treat its definition as an empirical question. Ultimately, the proof is in the pudding: we consider a representation to be good if it makes subsequent

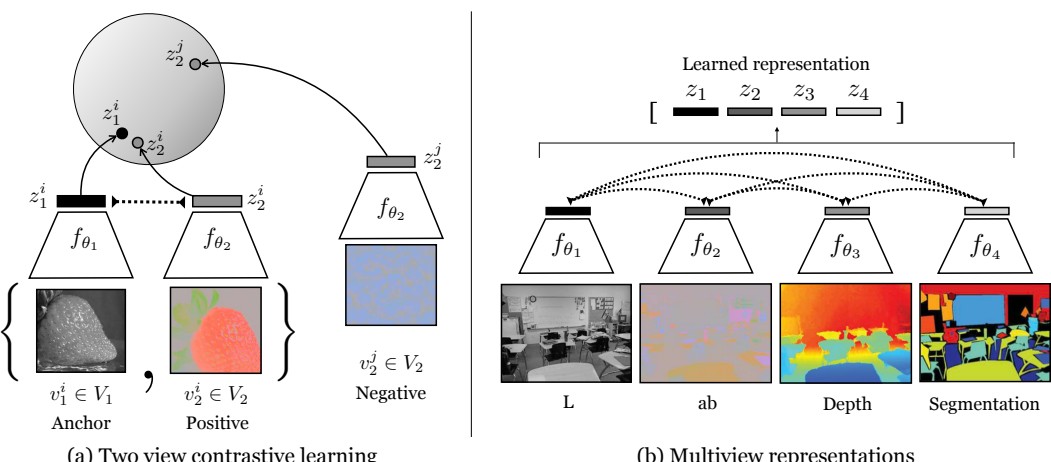

(a) Two view contrastive learning  (b) Multiview representations

Figure 1: (a) Given a pair of sensory views, a deep representation is learnt by bringing views of the *same* scene together in embedding space, while pushing views of *different* scenes apart. Here we show an example of learning from the luminance channel (L) of an image and the ab-color channel. The strawberry's L and ab channels embed to nearby points whereas the ab channel of a different image (a photo of blueberries) embeds to a far away point. (b) Example of a 4-view dataset (NYU RGBD (Nathan Silberman & Fergus, 2012)) and its learned representation. Dotted lines represent the contrastive objective. The encodings for each view may be concatenated to form the full representation of a scene.

problem solving easy, on tasks of human interest. For example, a useful representation of images might be a feature space in which it is easy to learn to recognize objects. We therefore evaluate our method by testing if the learned representations transfer well to standard semantic recognition tasks. On several benchmark tasks, our method achieves state of the art results, compared to other methods for self-supervised representation learning. We additionally find that the quality of the representation improves as a function of the number of views used for training. Finally, we compare the contrastive formulation of multiview learning to the recently popular approach of cross-view prediction, and find that in head-to-head comparisons, the contrastive approach learns stronger representations.

The core ideas that we build on: contrastive learning, mutual information maximization, and deep representation learning, are not new and have been explored in the literature on representation and multiview learning (Li et al., 2018; Xu et al., 2013; Arora et al., 2019). Our main contribution is to set up a framework to extend these ideas to any number of views, and to empirically study the factors that lead to success in this framework. A review of the related literature is given at the end of the paper, in Section 4. Fig. 1 gives a pictorial overview of our framework for the different learning tasks we consider in this paper, to learn representations across datasets with different sets of views. Our main contributions are:

- We apply contrastive learning to the multiview setting, attemping to maximize mutual information between representations of different views of the same scene (e.g., between different image channels, or different modalities).
- Our approach yields representations that outperform the state-of-the-art in self-supervised learning in head-to-head comparisons. For example, in the ImageNet linear readoff evaluation, we achieve $68.4\%$ top-1 accuracy, which is slightly above the state of the art concurrent work Bachman et al. (2019).
- We show that the contrastive objective is superior to cross-view prediction.
- We extend the framework to learn from *more than two* views, and show that the quality of the learned representation improves as number of views increase.
- We conduct controlled experiments to measure the effect of mutual information on representation quality.

## 2 METHOD

Our goal is to learn representations that capture information shared between multiple sensory views without human supervision. We start by reviewing previous predictive learning (or reconstruction-

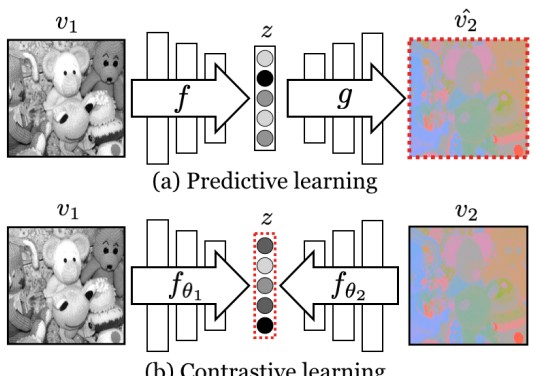

(a) Predictive learning

(b) Contrastive learning

Figure 2: Predictive Learning vs Contrastive Learning. Cross-view prediction (**Top**) learns latent representations that predict one view from another, with loss measured in the *output* space. Common prediction losses, such as the $\mathcal{L}_1$ and $\mathcal{L}_2$ norms, are *unstructured*, in the sense that they penalize each output dimension independently, perhaps leading to representations that do not capture all the shared information between the views. In contrastive learning (**Bottom**), representations are learnt by contrasting congruent and incongruent views, with loss measured in *representation* space. The red dotted outlines show where the loss function is applied.

based learning) methods, and then elaborate on contrastive learning within two views. We show connections to mutual information maximization and extend it to scenarios including more than two views. We consider a collection of $M$ views of the data, denoted as $V_1, \ldots, V_M$. For each view $V_i$, we denote $v_i$ as a random variable representing samples following $v_i \sim \mathcal{P}(V_i)$.

## 2.1 PREDICTIVE LEARNING

Let $V_1$ and $V_2$ represent two views of a dataset. For instance, $V_1$ might be the luminance of a particular image and $V_2$ the chrominance. We define the *predictive learning* setup as a deep nonlinear transformation from $v_1$ to $v_2$ through latent variables $z$, as shown in Fig. 2. Formally, $z = f(v_1)$ and $\hat{v_2} = g(z)$, where $f$ and $g$ represent the encoder and decoder respectively and $\hat{v_2}$ is the prediction of $v_2$ given $v_1$. The parameters of the encoder and decoder models are then trained using an objective function that tries to bring $\hat{v_2}$ "close to" $v_2$. Simple examples of such an objective include the $\mathcal{L}_1$ or $\mathcal{L}_2$ loss functions. Note that these objectives assume independence between each pixel or element of $v_2$ given $v_1$, i.e., $p(v_2|v_1) = \Pi_i p(v_{2i}|v_1)$, thereby reducing their ability to model correlations or complex structure. The predictive approach has been extensively used in representation learning, for example, colorization (Zhang et al., 2016; 2017) and predicting sound from vision (Owens et al., 2016).

## 2.2 CONTRASTIVE LEARNING WITH TWO VIEWS

The idea behind contrastive learning is to learn an embedding that separates (contrasts) samples from two different distributions. Given a dataset of $V_1$ and $V_2$ that consists of a collection of samples $\{v_1^i, v_2^i\}_{i=1}^N$, we consider contrasting congruent and incongruent pairs, i.e. samples from the joint distribution $x \sim p(v_1, v_2)$ or $x = \{v_1^i, v_2^i\}$, which we call *positives*, versus samples from the product of marginals, $y \sim p(v_1)p(v_2)$ or $y = \{v_1^i, v_2^j\}$, which we call *negatives*.

We learn a "critic" $h_\theta(\cdot)$ that is high for positives and low for negatives. Similar to recent setups for contrastive learning (Oord et al., 2018; Gutmann & Hyvärinen, 2010; Mnih & Kavukcuoglu, 2013), we train this function to correctly select a single positive sample $x$ out of a set $S = \{x, y_1, y_2, ..., y_k\}$ that contains $k$ negative samples:

$$\mathcal{L}_{contrast} = -\mathop{\mathbb{E}}_{S} \left[ \log \frac{h_\theta(x)}{h_\theta(x) + \sum_{i=1}^k h_\theta(y_i)} \right] \tag{1}$$

To construct $S$, we simply fix one view and enumerate positives and negatives from the other view, allowing us to rewrite the objective as:

$$\mathcal{L}_{contrast}^{V_1, V_2} = -\mathop{\mathbb{E}}_{\{v_1^1, v_2^1, ..., v_2^{k+1}\}} \left[ \log \frac{h_\theta(\{v_1^1, v_2^1\})}{\sum_{j=1}^{k+1} h_\theta(\{v_1^1, v_2^j\})} \right] \tag{2}$$

where $k$ is the number of negative samples $v_2^j$ for a given sample $v_1^1$. In practice, $k$ can be extremely large, and so directly minimizing Eq. 2 is infeasible. In Section 2.4, we show an approximation based on Noise Contrastive Estimation (Gutmann & Hyvärinen, 2010) that allows for tractable computation.

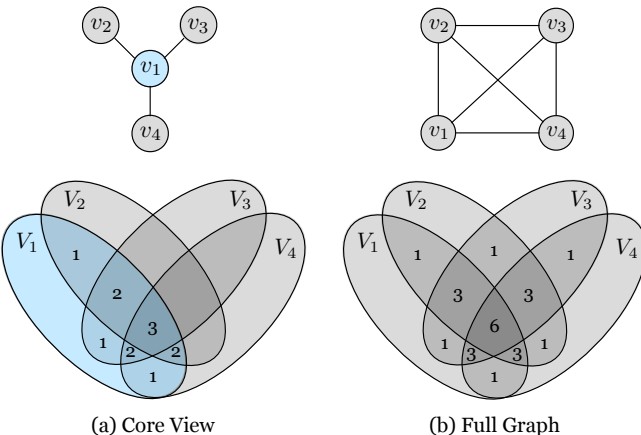

Figure 3: Graphical models and information diagrams (inf) associated with the core view and full graph paradigms, for the case of 4 views, which gives a total of 6 learning objectives. The numbers within the regions show how much "weight" the total loss places on each partition of information (i.e. how many of the 6 objectives that partition contributes to). A region with no number corresponds to 0 weight. For example, in the full graph case, the mutual information between all 4 views is considered in all 6 objectives, and hence is marked with the number 6.

(a) Core View      (b) Full Graph

**Implementing the critic**   We implement the critic $h_\theta(\cdot)$ as a neural network. To extract compact latent representations of $v_1$ and $v_2$, we employ two encoders $f_{\theta_1}(\cdot)$ and $f_{\theta_2}(\cdot)$ with parameters $\theta_1$ and $\theta_2$ respectively. The latent representions are extracted as $z_1 = f_{\theta_1}(v_1)$, $z_2 = f_{\theta_2}(v_2)$. On top of these features, the score is computed as the exponential of a bivariate function of $z_1$ and $z_2$, e.g., a bilinear function parameterized by $W_{12}$:

$$h_\theta(\{v_1, v_2\}) = e^{f_{\theta_1}(v_1)^T W_{12} f_{\theta_2}(v_2)} \tag{3}$$

Loss $\mathcal{L}_{contrast}^{V_1,V_2}$ in Eq. 2 treats view $V_1$ as anchor and enumerates over $V_2$. Symmetrically, we can get $\mathcal{L}_{contrast}^{V_2,V_1}$ by anchoring at $V_2$. We add them up as our two-view loss:

$$\mathcal{L}(V_1, V_2) = \mathcal{L}_{contrast}^{V_1,V_2} + \mathcal{L}_{contrast}^{V_2,V_1} \tag{4}$$

After the contrastive learning phase, we use the representation $z_1$, $z_2$, or the concatenation of both, $[z_1, z_2]$, depending on our paradigm. This process is visualized in Fig. 1.

**Connecting to mutual information**   The optimal critic $h_\theta^*$ is proportional to the density ratio between the joint distribution $p(z_1, z_2)$ and the product of marginals $p(z_1)p(z_2)$ (proof provided in Sec. C.1):

$$h_\theta^*(\{v_1, v_2\}) \quad \propto \quad \frac{p(z_1, z_2)}{p(z_1)p(z_2)} \propto \frac{p(z_1|z_2)}{p(z_1)} \tag{5}$$

This quantity is the pointwise mutual information, and its expectation, in Eq. 2, yields an estimator related to mutual information. A formal proof is given by Oord et al. (2018); Poole et al. (2019), which we recapitulate in Section C, showing that:

$$I(z_i; z_j) \geq \log(k) - \mathcal{L}_{contrast} \tag{6}$$

where, as above, $k$ is the number of negative pairs in sample set $S$. Hence minimizing the objective $\mathcal{L}$ maximizes the lower bound on the mutual information $I(z_i; z_j)$, which is bounded above by $I(v_i; v_j)$ by the data processing inequality. The dependency on $k$ also suggests that using more negative samples can lead to an improved representation; we show that this is indeed the case in Section A.1.2. We note that recent work (McAllester & Statos, 2018) shows that the bound in Eq. 6 can be very weak; and finding better estimators of mutual information is an important open problem.

## 2.3   Contrastive Learning with More than Two Views

We present more general formulations of Eq. 2 that can handle any number of views. We call them the "core view" and "full graph" paradigms, which offer different tradeoffs between efficiency and effectiveness. These formulations are visualized in Fig. 3.

Suppose we have a collection of $M$ views $V_1, \ldots, V_M$. The "core view" formulation sets apart one view that we want to optimize over, say $V_1$, and builds pair-wise representations between $V_1$ and each

other view $V_j, j > 1$, by optimizing the sum of a set of pair-wise objectives:

$$\mathcal{L}_C = \sum_{j=2}^{M} \mathcal{L}(V_1, V_j) \tag{7}$$

A second, more general formulation is the "full graph" where we consider all pairs $(i, j), i \neq j$, and build $\binom{n}{2}$ relationships in all. By involving all pairs, the objective function that we optimize is:

$$\mathcal{L}_F = \sum_{1 \leq i < j \leq M} \mathcal{L}(V_i, V_j) \tag{8}$$

Both these formulations have the effect that information is prioritized in proportion to the number of views that share that information. This can be seen in the information diagrams visualized in Fig. 3. The number in each partition of the diagram indicates how many of the pairwise objectives, $\mathcal{L}(V_i, V_j)$, that partition contributes to. Under both the core view and full graph objectives, a factor, like "presence of dog", that is common to all views will be preferred over a factor that affects fewer views, such as "depth sensor noise".

The computational cost of the bivariate score function in the full graph formulation is combinatorial in the number of views. However, it is clear from Fig. 3 that this enables the full graph formulation to capture more information between different views, which may prove useful for downstream tasks. For example, the mutual information between $V_2$ and $V_3$ or $V_2$ and $V_4$ is completely ignored in the core view paradigm (as shown by a 0 count in the information diagram).

## 2.4 Approximating the Softmax Distribution with Noise-Contrastive Estimation

Better representations using $\mathcal{L}_{contrast}^{V_1, V_2}$ in Eq. 2 are learnt by using many negative samples. However, computing the full softmax loss is prohibitively expensive for large $N$. We alleviate computational load by using Noise-Contrastive Estimation (NCE) (Gutmann & Hyvärinen, 2010) to approximate the full softmax in Eqn. 2, as has also been used in Mnih & Kavukcuoglu (2013)[1].

Given an anchor $v_1^i$ from $V_1$, the probablity that an atom $v_2 \in \{v_2^j | j = 1, 2, ..., N\}$ from $V_2$ is the best match of $v_1^i$, using the score $h_\theta$ is given by:

$$p(v_2 | v_1^i) = \frac{h_\theta(\{v_1^i, v_2\})}{\sum_{j=1}^{N} h_\theta(\{v_1^i, v_2^j\})} \tag{9}$$

where the normalization factor $Z = \sum_{j=1}^{N} h_\theta(\{v_1^i, v_2^j\})$ is expensive to compute for large $N$. Here we use $h_\theta(\{v_1, v_2\}) = \exp(f_{\theta_1}(v_1)^T W_{12} f_{\theta_2}(v_2)/\tau)$, where $\tau$ modulates the distribution.

Noise-Contrastive Estimation (Gutmann & Hyvärinen, 2010) (NCE) is an effective way to estimate *unnormalized* statistical models. NCE fits a density model $p$ to data distributed as (unknown) distribution $p_d$, by using a binary classifier to distinguish it from noise samples distributed as $p_n$. To learn $p(v_2 | v_1^i)$, we use a binary classifier, which treats $v_2$ as the data sample when given $v_1^i$. The noise distribution $p_n(\cdot | v_1^i)$ we choose here is a uniform distribution over all atoms from $V_2$, i.e., $p_n(\cdot | v_1^i) = 1/N$. If we sample $m$ noise samples to pair with each data sample, the posterior probability that a given atom $v_2$ comes from the data distribution is:

$$P(D = 1 | v_2; v_1^i) = \frac{p_d(v_2 | v_1^i)}{p_d(v_2 | v_1^i) + m p_n(v_2 | v_1^i)} \tag{10}$$

and we estimate this probability by replacing $p_d(v_2 | v_1^i)$ with our *unnormalized* model distribution $h_\theta(v_1^i, v_2)$. Minimizing the negative log-posterior probability of correct labels $D$ over data and noise samples yields our final objective, which is the NCE-based approximation of Eq. 2 ($\hat{p}$ is the empirical data distribution):

$$L_{NCE} = - \mathbb{E}_{v_1^i \sim \hat{p}(v_1)} \left\{ \mathbb{E}_{v_2 \sim \hat{p}(\cdot | v_1^i)} \left[ \log(P(D = 1 | v_2; v_1^i)) \right] + m \mathbb{E}_{v_2 \sim p_n(\cdot | v_1^i)} \left[ \log(P(D = 0 | v_2; v_1^i)) \right] \right\} \tag{11}$$

---

[1]Confusingly, the literature has previously referred to Eqn. 2 as "InfoNCE" (Oord et al., 2018). Our NCE approximation *does not* refer to the allusion to NCE in the name "InfoNCE". Rather we are here describing an NCE approximation to the "InfoNCE" softmax objective.

**Memory bank.** Following Wu et al. (2018), we maintain a memory bank to store latent features for each training sample. Therefore, we can efficiently retrieve $m$ noise samples from the memory bank to pair with each positive sample without recomputing their features. The memory bank is dynamically updated with features computed on the fly.

An alternative to the NCE based approximation above, is to simply do $m+1$ way softmax classification with $m$ noise samples retrieved from the memory bank. We note that CPC (Oord et al., 2018) and Deep InfoMax (Hjelm et al., 2019) use this $m + 1$ way softmax classification as their ultimate contrastive loss rather than the NCE-based contrastive loss in Eq. 11 (but note that CPC refers to the $m + 1$ approximation as also "based on NCE"). Empirically we have found that the $m + 1$-way softmax classification approach performed worse than our NCE-based approximation, given the same number of noise samples.

## 3 EXPERIMENTS

We extensively evaluate Contrastive Multiview Coding (CMC) on a number of datasets and tasks. We evaluate on two established image representation learning benchmarks: Imagenet and STL-10 (See A.1.1). We further validate our framework on video representation learning tasks (See A.2), where we use image and optical flow modalities, as the two views that are jointly learned. The last set of experiments extends our CMC framework to more than two views and provides empirical evidence of it's effectiveness.

### 3.1 BENCHMARKING CMC ON IMAGENET

Following Zhang et al. (2016), we evaluate task generalization of the learned representation by training 1000-way *linear* classifiers on top of different layers.

**Setup.** Given a dataset of RGB images, we convert them to the *Lab* image color space, and split each image into *L* and *ab* channels, as originally proposed in SplitBrain autoencoders (Zhang et al., 2017). During contrastive learning, L and ab from the same image are treated as the positive pair, and ab channels from other randomly selected images are treated as a negative pair (for a given L). Each split represents a view of the orginal image and is passed through a separate encoder. As in SplitBrain, we design these two encoders by evenly splitting a given deep network, such as AlexNet (Krizhevsky et al., 2012), into sub-networks across the channel dimension. By concatenating representations layer-wise from these two encoders, we achieve the final representation of an input image. As proposed by previous literature (Oord et al., 2018; Hjelm et al., 2019; Arora et al., 2019), the quality of such a representation is evaluated by freezing the weights of encoder and training linear or non-linear classifiers on top of each layer.

To compare with other methods, we adopt standard AlexNet and split it into two encoders. Because of splitting, each layer only connects to half of the neurons in the previous layer, and therefore the number of parameters in our model halves. We remove local response layer and add batch normalization to each layer. For the memory-based CMC model, we adopt ideas from Wu et al. (2018) for computing and storing a memory. We retrieve 4096 negative pairs from the memory bank to contrast each positive pair. The training details are present in Sec. D.2.

Table 1 shows the results of comparing the CMC against other models, both predictive and contrastive. Our CMC is the best among all these methods; futhermore CMC tends to perform better at higher convolutional layers, similar to another contrasting-based model Inst-Dis (Wu et al., 2018).

**CMC with ResNets.** We verify the scalability of CMC with larger networks such as ResNets He et al. (2016). Here we do not split ResNets, rather we use ResNet-50, ResNet-101 or ResNet-50 x2 to encoder each of the two views (*L* and *ab*). The results are shown in Table 2, where ResNet-50, ResNet-101, and ResNet-50 x2 achieve 64.1%, 65.0%, and 68.4% top-1 accuracies, respectively. To our best knowledge, 68.4% on ImageNet is the highest published accuracy ever achieved by self-supervsied/unsupervised methods (We note that a concurrent work AMDIM Bachman et al. (2019) achieves similar results).

| Method | ImageNet Classification Accuracy | | | | |
|---|---|---|---|---|---|
| | conv1 | conv2 | conv3 | conv4 | conv5 |
| ImageNet-Labels | 19.3 | 36.3 | 44.2 | 48.3 | 50.5 |
| Random | 11.6 | 17.1 | 16.9 | 16.3 | 14.1 |
| Data-Init (Krähenbühl et al., 2015) | 17.5 | 23.0 | 24.5 | 23.2 | 20.6 |
| Context (Doersch et al., 2015) | 16.2 | 23.3 | 30.2 | 31.7 | 29.6 |
| Colorization (Zhang et al., 2016) | 13.1 | 24.8 | 31.0 | 32.6 | 31.8 |
| Jigsaw (Noroozi & Favaro, 2016) | 19.2 | 30.1 | 34.7 | 33.9 | 28.3 |
| BiGAN (Donahue et al., 2017) | 17.7 | 24.5 | 31.0 | 29.9 | 28.0 |
| SplitBrain[†] (Zhang et al., 2017) | 17.7 | 29.3 | 35.4 | 35.2 | 32.8 |
| Counting (Noroozi et al., 2017) | 18.0 | 30.6 | 34.3 | 32.5 | 25.7 |
| Inst-Dis (Wu et al., 2018) | 16.8 | 26.5 | 31.8 | 34.1 | 35.6 |
| RotNet (Gidaris et al., 2018) | 18.8 | 31.7 | 38.7 | 38.2 | 36.5 |
| DeepCluster (Caron et al., 2018) | 12.9 | 29.2 | 38.2 | 39.8 | 36.1 |
| AET (Zhang et al., 2019) | **19.3** | 32.8 | **40.6** | 39.7 | 37.7 |
| CMC | 18.4 | **33.5** | 38.1 | **40.4** | **42.6** |

Table 1: Top-1 classification accuracy on 1000 classes of ImageNet Deng et al. (2009) with single crop. We compare our CMC method with other unsupervised representation learning approaches by training 1000-way logistic regression classifiers on top of the feature maps of each layer, as proposed by Zhang et al. (2016). Methods marked with [†] only have half the number of parameters compared to others, because of splitting.

| Accuracy (%) | ResNet-50 | ResNet-101 | ResNet-50 x2 |
|---|---|---|---|
| Top-1 | 64.1 | 65.0 | 68.4 |

Table 2: Single crop top-1 classification accuracy on ImageNet. We evaluate CMC with ResNet-50, ResNet-101, or ResNet-50 x2 as encoder for *each* of the two views (*L* and *ab*).

## 3.2 DOES REPRESENTATION QUALITY IMPROVE AS NUMBER OF VIEWS INCREASES?

We further extend our CMC learning framework to multiview scenarios. We experiment on the NYU-Depth-V2 (Nathan Silberman & Fergus, 2012) dataset which consists of 1449 labeled images. We focus more on understanding the behavior and effectiveness of CMC rather than competing with the current state-of-the-arts. The views we consider are: luminance (L channel), chrominance (ab channel), depth, surface normal (Eigen & Fergus, 2015), and semantic labels.

**Setup.** To extract features from each view, we use a neural network with 5 convolutional layers, and 1 fully connected layer. As the size of the dataset is relatively small, we adopt the sub-patch based contrastive objective (see B) to increase the number of negative pairs. Patches with a size of $128 \times 128$ are randomly cropped from the original images for contrastive learning (from images of size $480 \times 640$). For downstream tasks, we discard the fully connected layers and evaluate using the convolutional layers as a representation.

To measure the quality of the learned representation, we consider the task of predicting semantic labels from the representation of $L$. We follow the *core view paradigm* and use $L$ are the core view, thus learning a set of representations on $L$ by contrasting different views with $L$. A UNet style architecture (Ronneberger et al., 2015) is utilized to perform the segmentation task. Contrastive training is performed on the above architecture that is equivalent of the UNet's encoder. After contrastive training is completed, we initialize the encoder weights of the UNet from the $L$ encoder (which are equivalent architectures) and keep them frozen. Only the decoder is trained during this finetuning stage.

Since we use the patch-based contrastive loss, in the 1 view setting case, CMC coincides with DIM (Hjelm et al., 2019). The 2-4 view cases contrast L with ab, and then sequentially add depth and surface normals. The semantic labeling results are measured by mean IoU over all classes and pixel accuracy, shown in Fig. 4. We see that the performance steadily improves as new views are added. We have tested different orders of adding the views, and they all follow a similar pattern.

We also compare CMC with two baselines. First, we randomly initialize and freeze the encoder, and we call this the *Random* baseline; it serves as a lower bound on the quality since the representation is just a random projection. Rather than freezing the randomly initialized encoder, we could train it

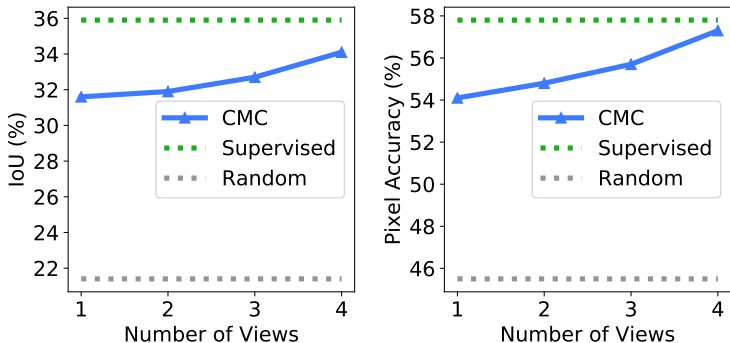

Figure 4: We show the Intersection over Union (IoU) (left) and Pixel Accuracy (right) for the NYU-Depth-V2 dataset, as CMC is trained with increasingly more views from 1 to 4. As more views are added, both these metrics steadily increase. The views are (in order of inclusion): L, ab, depth and surface normals.

|  | Pixel Accuracy (%) | mIoU (%) |
|---|---|---|
| Random | 45.5 | 21.4 |
| CMC (core-view) | 57.1 | 34.1 |
| CMC (full-graph) | 57.0 | 34.4 |
| Supervised | **57.8** | **35.9** |

Table 3: Results on the task of predicting semantic labels from **L channel** representation which is learnt using the patch-based contrastive loss and all 4 views. We compare CMC with *Random* and *Supervised* baselines, which serve as lower and upper bounds respectively. Th core-view paradigm refers to Fig. 3(a), and full-view Fig. 3(b).

jointly with the decoder. This end-to-end *Supervised* baseline serves as an upper bound. The results are presented in Table 3, which shows our CMC produces high quality feature maps even though it's unaware of the downstream task.

## 3.3    PREDICTIVE LEARNING VS. CONTRASTIVE LEARNING

While experiments in section 3.1 show that contrastive learning outperforms predictive learning (Zhang et al., 2017) in the context of Lab color space, it's unclear whether such an advantage is due to the natural inductive bias of the task itself. To further understand this, we go beyond chrominance (ab), and try to answer this question when geometry or semantic labels are present.

We consider three view pairs on the NYU-Depth dataset: (1) L and depth, (2) L and surface normals, and (3) L and segmentation map. For each of them, we train two identical encoders for L, one using contrastive learning and the other with predictive learning. We then evaluate the representation quality by training a linear classifier on top of these encoders on the STL-10 dataset.

| Views | Accuracy on STL-10 (%) | |
|---|---|---|
|  | Predictive | Contrastive |
| L, Depth | 55.5 | **58.3** |
| L, Normal | 58.4 | **60.1** |
| L, Seg. Map | 57.7 | **59.2** |
| Random | 25.2 | |
| Supervised | 65.1 | |

Table 4: We compare predictive learning with contrastive learning by evaluating the learned encoder on unseen dataset and task. The contrastive learning framework consistently outperforms predictive learning.

The comparison results are shown in Table 4, which shows that contrastive learning consistently outperforms predictive learning in this scenario where both the task and the dataset are unknown. We also include "random" and "supervised" baselines similar to that in previous sections. Though in the unsupervised stage we only use 1.3K images from a dataset much different from the target dataset STL-10, the object recognition accuracy is close to the supervised method, which uses an end-to-end deep network directly trained on STL-10.

Given two views $V_1$ and $V_2$ of the data, the predictive learning approach approximately models $p(v_2|v_1)$. Furthermore, losses used typically for predictive learning, such as pixel-wise reconstruction losses usually impose an independence assumption on the modeling: $p(v_2|v_1) \approx \Pi_i p(v_{2i}|v_1)$. On the other hand, the contrastive learning approach by construction does not assume conditional

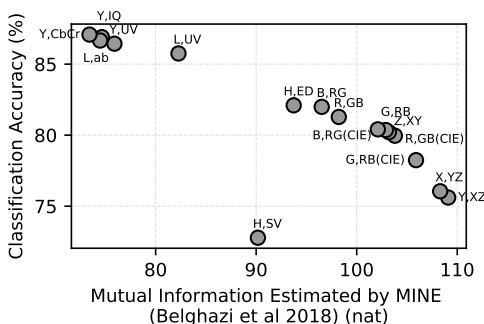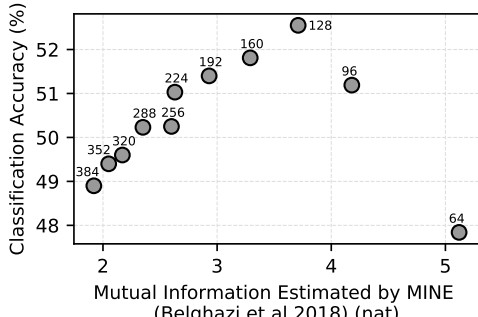

Figure 5: How does mutual information between views relate to representation quality? (Left) Classification accuracy against estimated MI between channels of different color spaces; (Right) Classification accuracy vs estimated MI between patches at different distances (distance in pixels is denoted next to each data point). MI estimated using MINE (Belghazi et al., 2018).

independence across dimensions of $v_2$. In addition, the use of random jittering and cropping between views allows the contrastive learning approach to benefit from spatial co-occurrence (contrasting in space) in addition to contrasting across views. We conjecture that these are two reasons for the superior performance of contrastive learning approaches over predictive learning.

### 3.4 HOW DOES MUTUAL INFORMATION AFFECT REPRESENTATION QUALITY?

Given a fixed set of views, CMC aims to maximize the mutual information between representations of these views. We have found that maximizing information in this way indeed results in strong representations, but it would be incorrect to infer that information maximization (infomax) is the key to good representation learning. In fact, this paper argues for precisely the opposite idea: that cross-view representation learning is effective because it results in a kind of information minimization, *discarding* nuisance factors that are not shared between the views.

The resolution to this apparent dilemma is that we want to maximize the "good" information – the *signal* – in our representations, while minimizing the "bad" information – the *noise*. The idea behind CMC is that this can be achieved by doing infomax learning on two views that share signal but have independent noise. This suggests a "Goldilocks principle": a good collection of views is one that shares some information but not too much. Here we test this hypothesis on two domains: learning representations on images with different colorspaces forming the two views; and learning representations on pairs of patches extracted from an image, separated by varying spatial distance.

In patch experiments we randomly crop two RGB patches of size 64x64 from the same image as two views. Their relative position is fixed. Namely, the two patches always starts at position $(x, y)$ and $(x + d, y + d)$ with $(x, y)$ being randomly sampled. While varying the distance $d$, we start from 64 to avoid overlapping. There is a possible bias that with an image of relatively small size (e.g., 512x512), a large $d$ (e.g., 384) will always push these two patches around boundary. To minimize this bias, we use high resolution images (e.g. $2k$) from DIV2K (Agustsson & Timofte, 2017) dataset.

Fig. 5 shows the results of these experiments. The left plot shows the result of learning representations on different colorspaces (splitting each colorspace into two views, such as (L, ab), (R, GB) etc). We then use the MINE estimator Belghazi et al. (2018) to estimate the mutual information between the views. We measure representation quality by training a linear classifier on the learned representations on the STL-10 dataset Coates et al. (2011). The plots clearly show that using colorspaces with minimal mutual information give the best downstream accuracy (For the outlier HSV in this plot, we conjecture the representation quality is harmed by the periodicity of H. Note that the H in HED is not periodic.). On the other hand, the story is more nuanced for representations learned between patches at different offsets from each other (Fig. 5, right). Here we see that views with too little or too much MI perform worse; a sweet spot in the middle exists which gives the best representation. That there exists such a sweet spot should be expected. If two views share *no* information, then, in principle, there is no incentive for CMC to learn anything. If two views share all their information, no

nuisances are discarded and we arrive back at something akin to an autoencoder or generative model, that simply tries to represent all the bits in the multiview data.

These experiments demonstrate that the relationship between mutual information and representation quality is meaningful but not direct. Selecting optimal views, which just share relevant signal, may be a fruitful direction for future research.

## 4 RELATED WORK

Unsupervised representation learning is about learning transformations of the data that make subsequent problem solving easier (Bengio et al., 2013). This field has a long history, starting with classical methods with well established algorithms, such as principal components analysis (PCA (Jolliffe, 2011)) and independent components analysis (ICA (Hyvärinen et al., 2004)). These methods tend to learn representations that focus on low-level variations in the data, which are not very useful from the perspective of downstream tasks such as object recognition.

Representations better suited to such tasks have been learnt using deep neural networks, starting with seminal techniques such as Boltzmann machines (Smolensky, 1986; Salakhutdinov & Hinton, 2009), autoencoders (Hinton & Salakhutdinov, 2006), variational autoencoders (Kingma & Welling, 2013), generative adversarial networks (Goodfellow et al., 2014) and autoregressive models (Oord et al., 2016). Numerous other works exist, for a review see (Bengio et al., 2013). A powerful family of models for unsupervised representations are collected under the umbrella of "self-supervised" learning (Sa, 2004; Zhang et al., 2017; 2016; Isola et al., 2015; Wang & Gupta, 2015; Pathak et al., 2016; Zhang et al., 2019). In these models, an input $X$ to the model is transformed into an output $\hat{X}$, which is supposed to be close to another signal $Y$, which itself is related to $X$ in some meaningful way. Examples of such $X/Y$ pairs are: luminance and chrominance color channels of an image (Zhang et al., 2017), patches from a single image (Oord et al., 2018), modalities such as vision and sound (Owens et al., 2016) or the frames of a video (Wang & Gupta, 2015). Clearly, such examples are numerous in the world, and provides us with nearly infinite amounts of training data: this is one of the appeals of this paradigm. Time contrastive networks (Sermanet et al., 2017) use a triplet loss framework to learn representations from aligned video sequences of the same scene, taken by different video cameras. Closely related to self-supervised learning is the idea of multi-view learning, which is a general term involving many different approaches such as co-training (Blum & Mitchell, 1998), multi-kernel learning (Cortes et al., 2009) and metric learning (Bellet et al., 2012; Zhuang et al., 2019); for comprehensive surveys please see (Xu et al., 2013; Li et al., 2018). Nearly all existing works have dealt with one or two views such as video or image/sound. However, in many situations, many more views are available to provide training signals for any representation.

The objective functions used to train deep learning based representations in many of the above methods are either reconstruction-based loss functions such as Euclidean losses in different norms e.g. (Isola et al., 2017), adversarial loss functions (Goodfellow et al., 2014) that learn the loss in addition to the representation, or contrastive losses e.g. (Gutmann & Hyvärinen, 2010; Hjelm et al., 2019; Oord et al., 2018; Arora et al., 2019; Hénaff et al., 2019) that take advantage of the co-occurence of multiple views. Another recently introduced novel objective function is instance discrimination (Wu et al., 2018). In this work, we compare the two most commonly used objectives: predictive and contrastive. The prior works most similar to our own (and inspirational to us) are Contrastive Predictive Coding (CPC) (Oord et al., 2018) and Deep InfoMax (Hénaff et al., 2019). These two methods, like ours, learn representations by contrasting between congruent and incongruent representations of a scene, and are motivated as forms of infomax learning. CPC learns from two views – the past and future – and is applicable to sequential data. Deep Infomax (Hjelm et al., 2019) considers the two views to be the input to a neural network and its output. These two methods share the same mathematical objective, but differ in the definition of the views. Our technical method is also highly related, but differs in the following ways: we extend the objective to the case of *more than two* views; and we use a loss function which more closely follows the original method of noise contrastive estimation (Gutmann & Hyvärinen, 2010) (See details in Section 2.4). Although CPC, Deep InfoMax, and the present paper are all very similar at the mathematical level, they each explore a different set of view definitions, architectures, and application settings, and each contributes its own unique empirical investigation of this paradigm of representation learning.

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

## A  ADDITIONAL EXPERIMENTS

### A.1  CMC ON IMAGES

Given a dataset of RGB images, we convert them to the *Lab* image color space, and split each image into *L* and *ab* channels, as originally proposed in SplitBrain autoencoders (Zhang et al., 2017). During contrastive learning, L and ab from the same image are treated as the positive pair, and ab channels from other randomly selected images are treated as a negative pair (for a given L). Each split represents a view of the orginal image and is passed through a seprate encoder. This corresponds to the "full graph" model of Eq. 8 with L and ab channels as the two views. As in SplitBrain, we design these two encoders by evenly splitting a given deep network, such as AlexNet (Krizhevsky et al., 2012), into sub-networks across the channel dimension. By concatenating representations layer-wise from these two encoders, we achieve the final representation of an input image. As proposed by previous literature (Oord et al., 2018; Hjelm et al., 2019; Arora et al., 2019), the quality of such a representation is evaluated by freezing the weights of encoder and training linear or non-linear classifiers on top of each layer.

### A.1.1 STL-10

STL-10 (Coates et al., 2011) is an image recognition dataset designed for developing unsupervised or self-supervised learning algorithms. It consists of 100000 unlabeled training $96 \times 96$ RGB image samples and 500 labeled samples for each of the 10 classes.

**Setup.** We adopt the same data augmentation strategy and network architecture as those in DIM (Hjelm et al., 2019). A variant of AlexNet takes as input $64 \times 64$ images, which are randomly cropped and horizontally flipped from the original $96 \times 96$ size images. For a fair comparison with DIM, we also train our model in a patch-based contrastive fashion during unsupervised pre-training. With the weights of the pre-trained encoder frozen, a two-layer fully connected network with 200 hidden units is trained on top of different layers for 100 epochs to perform 10-way classification. We also investigated the strided crop strategy of CPC (Oord et al., 2018). Fixed sized overlapping patches of size $16 \times 16$ with an overlap of $8$ pixels are cropped and fed into the network separately. This ensures that features of one patch contain minimal information from neighbouring patches; and increases the available number of negative pairs for the contrastive loss. Additionally, we include NCE-based contrastive training and linear classifier evaluation.

**Comparison.** We compare CMC with the state of the art unsupervised methods in Table 5. Three columns are shown: the conv5 and fc7 columns use respectively these layers of AlexNet as the encoder (again remembering that we split across channels for L and ab views). For these two columns we can compare against the all methods except CPC, since CPC does not report these numbers in their paper (Hjelm et al., 2019). In the Strided Crop setup, we only compare against the approaches that use contrastive learning, DIM and CPC, since this method was only used by those works. We note that in Table 5 for all the methods except SplitBrain, we report numbers are shown in the original paper. For SplitBrain, we reimplemented their model faithfully and report numbers based on our reimplementation (we verified the accuracy of our SplitBrain code by the fact that we get very similar results with our reimpementation as in the original paper (Zhang et al., 2017) for ImageNet experiments, see below).

The family of contrastive learning methods, such as DIM, CPC, and CMC, achieve higher classification accuracy than other methods such as SplitBrain that use predictive learning; or BiGAN that use adversarial learning. CMC significantly outperforms DIM and CPC in all cases. We hypothesize that this outperformance results from the modeling of cross-view mutual information, where view-specific noisy details are discarded. Another head-to-head comparison happens between CMC and SplitBrain, both of which modeling images as seprated L and ab streams; we achieve a nearly $8\%$ absolute improvement for conv5 and $17\%$ improvement for fc5. Finally, we notice that the predictive learning methods suffer from a big drop in performance when the encoding layer is switched from conv5 to fc7. On the other hand, the contrastive learning approaches are much more stable across layers, suggesting that the mutual information maximization paradigm learns more semantically meaningful representations shared by the different views. From a practical perspective, this is a significant advantage as the selection of specific layers should ideally not change downstream performance by too much.

In this experiments we used AlexNet as backbone. Switching to more powerful networks such as ResNets is likely to further improve the representation quality.

### A.1.2 IMAGENET

ImageNet (Deng et al., 2009) consists of 1000 image classes and is frequently considered as a testbed for unsupervised representation learning algorithms.

**Effect of the number of negative samples.** We investigate the relationship between the number of negative pairs $m$ in NCE-based loss and the downstream classification accuracy on a randomly chosen subset of 100 classes of Imagenet (the same set of classes is used for any number of negative pairs). We train a 100-way linear classifier using CMC pre-trained features with varying number of negative pairs, starting from 64 pairs upto 8192 (in multiples of 2). Fig. 6 shows that the accuracy of the resulting classifier steadily increases but saturates at around $60.3\%$ with $m = 4096$ samples. AlexNet is used in this study.

| Method | classifier | conv5 | fc7 | Strided Crop |
|---|---|---|---|---|
| AE | | 62.19 | 55.78 | - |
| NAT (Bojanowski & Joulin, 2017) | MLP | 64.32 | 61.43 | - |
| BiGAN (Donahue et al., 2017) | | 71.53 | 67.18 | - |
| SplitBrain[†] (Zhang et al., 2017) | | 72.35 | 63.15 | - |
| DIM (Hjelm et al., 2019) | MLP | 72.57 | 70.00 | 78.21 |
| CPC (Oord et al., 2018) | | - | - | 77.81 |
| CMC[†] (Patch) | Linear | 76.65 | 79.25 | 82.58 |
| CMC[†] (Patch) | MLP | 80.14 | 80.11 | **83.43** |
| CMC[†] (NCE) | Linear | 83.28 | 86.66 | - |
| CMC[†] (NCE) | MLP | **84.64** | **86.88** | - |
| Supervised | | | 68.70 | |

Table 5: Classification accuracies on STL-10 by using a two layer MLP as classifier for evaluating the representations learned by a small **AlexNet**. For all methods we compare against, we include the numbers that are reported in the DIM (Hjelm et al., 2019) paper, except for SplitBrain, which is our reimplementation. Methods marked with [†] have half the number of parameters because of splitting.

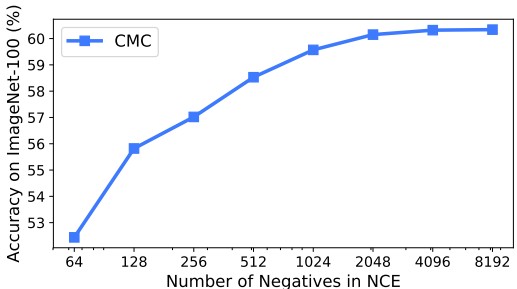

Figure 6: We plot the number of negative examples $m$ in NCE-based contrastive loss against the accuracy for 100 randomly chosen classes of Imagenet 100. It is seen that the accuracy steadily increases with $m$.

## A.2 CMC ON VIDEOS

We apply CMC on videos by drawing insight from the two-streams hypothesis (Schneider, 1969; Goodale & Milner, 1992), which posits that human visual cortex consists of two distinct processing streams: the ventral stream, which performs object recognition, and the dorsal stream, which processes motion. In our formulation, given an image $i_t$ that is a frame centered at time $t$, the ventral stream associates it with a neighbouring frame $i_{t+k}$, while the dorsal stream connects it to optical flow $f_t$ centered at $t$. Therefore, we extract $i_t$, $i_{t+k}$ and $f_t$ from two modalities as three views of a video; for optical flow we use the TV-L1 algorithm (Zach et al., 2007). Two separate contrastive learning objectives are built within the ventral stream $(i_t, i_{t+k})$ and within the dorsal stream $(i_t, f_t)$. For the ventral stream, the negative sample for $i_t$ is chosen as a random frame from another randomly chosen video; for the dorsal stream, the negative sample for $i_t$ is chosen as the flow corresponding to a random frame in another randomly chosen video.

**Pre-training.** We train CMC on UCF101 (Soomro et al., 2012) and use two CaffeNets (Krizhevsky et al., 2012) for extracting features from images and optical flows, respectively. In our implementation, $f_t$ represents 10 continuous flow frames centered at $t$. We use batch size of 128 and contrast each positive pair with 127 negative pairs. CMC is trained with Adam for 300 epochs, with an initial learning rate of 0.001 which is decayed by a factor of 5 after 200 and 250 epochs.

**Action recognition.** We apply the learn representation to the task of action recognition. The spatial network from (Simonyan & Zisserman, 2014) is a well-established paradigm for evaluating pre-trained RGB network on action recognition task. We follow the same spirit and evaluate the transferability of our RGB CaffeNet on UCF101 and HMDB51 datasets. We initialize the action recognition CaffeNet up to conv5 using the weights from the pre-trained RGB CaffeNet. The averaged accuracy over three splits is present in Table 6. Unifying both ventral and dorsal streams during pre-training produces higher accuracy for downstream recognition than using only single stream. Increasing the number

| Method | # of Views | UCF-101 | HMDB-51 |
|---|---|---|---|
| Random | - | 48.2 | 19.5 |
| ImageNet | - | 67.7 | 28.0 |
| VGAN* (Vondrick et al., 2016) | 2 | 52.1 | - |
| LT-Motion* (Luo et al., 2017) | 2 | 53.0 | - |
| TempCoh (Mobahi et al., 2009) | 1 | 45.4 | 15.9 |
| Shuffle and Learn (Misra et al., 2016) | 1 | 50.2 | 18.1 |
| Geometry (Gan et al., 2018) | 2 | 55.1 | 23.3 |
| OPN (Lee et al., 2017) | 1 | 56.3 | 22.1 |
| ST Order (Buchler et al., 2018) | 1 | 58.6 | 25.0 |
| Cross and Learn (Sayed et al., 2018) | 2 | 58.7 | **27.2** |
| CMC (V) | 2 | 55.3 | - |
| CMC (D) | 2 | 57.1 | - |
| CMC (V+D) | 3 | **59.1** | 26.7 |

Table 6: Test accuracy (%) on UCF-101 which evaluates *task* transferability and on HMDB-51 which evaluates *task* and *dataset* transferability. Most methods either use single RGB view or additional optical flow view, while VGAN explores sound as the second view. * indicates different network architecture.

| | Metric (%) | L | ab | Depth | Normal |
|---|---|---|---|---|---|
| Random | mIoU | 21.4 | 15.6 | 30.1 | 29.5 |
| | pix. acc. | 45.5 | 37.7 | 51.1 | 50.5 |
| CMC | mIoU | 34.4 | 26.1 | 39.2 | 37.8 |
| | pix. acc. | 57.0 | 49.6 | **59.4** | 57.8 |
| Supervised | mIoU | **35.9** | **29.6** | **41.0** | **41.5** |
| | pix. acc. | **57.8** | **52.6** | 59.1 | **59.6** |

Table 7: Performance on the task of using single view $v$ to predict the semantic labels, where $v$ can be L, ab, depth or surface normal. Our CMC framework improves the quality of unsupervised representations towards that of supervised ones, for all of views investigated. This uses the full-graph paradigm Fig. **??**(b).

of views of the data from 2 to 3 (using both streams instead of one) provides a boost for UCF-101. Furthermore, on UCF-101, we outperform all other methods; and on HMDB-51, CMC is second-best in performance.

### A.3 IS CMC IMPROVING ALL VIEWS?

A desirable unsupervised representation learning algorithm operating on multiple views or modalities should improve the quality of representations for all views. We therefore investigate our CMC framwork beyond L channel. To treat all views fairly, we train these encoders following the *full graph paradigm*, where each view is contrasted with all other views.

We evaluate the representation of each view $v$ by predicting the semantic labels from only the representation of $v$, where $v$ is L, ab, depth or surface normals. This uses the full-graph paradigm. As in the previous section, we compare CMC with *Random* and *Supervised* baselines. As shown in Table 7, the performance of the representations learned by CMC using full-graph significantly outperforms that of randomly projected representations, and approaches the performance of the fully supervised representations. Furthermore, the full-graph representation provides a good representation learnt for all views, showing the importance of capturing different types of mutual information across views.

## B CONTRASTING SUB-PATCHES

Instead of contrasting features from the last layer, patch-based method (Hjelm et al., 2019) contrasts feature from the last layer with features from previous layers, hence increasing the number of negative pairs. For instance, we use features from the last layer of $f_{\theta_1}$ to contrast with feature points from feature maps produced by the first several conv layers of $f_{\theta_2}$. This is equivalent to contrast between global patch from one view with local patches from the other view. In this fashion, we directly

perform $m + 1$ way softmax classification, the same as (Oord et al., 2018; Hjelm et al., 2019) for a fair comparison in Sec. A.1.1.

Such patch-based contrastive loss is computed within each mini-batch and does not require a memory bank. Therefore, deploying it in parallel training schemes is easy and flexible. However, patch-based contrastive loss usually yields suboptimal results compared to NCE-based contrastive loss, according to our experiments.

## C  PROOFS

We prove that: (a) the optimal score function $h_\theta^*(\{v_1, v_2\})$ is proportional to density ratio between the joint distribution $p(v_1, v_2)$ and product of marginals $p(v_1)p(v_2)$, as shown in Eq. 5; (b) Minimizing the contrastive loss $\mathcal{L}_{contrast}$ maxmizes a lower bound on the mutual information between two views, as shown in Eq. 6

We will use the most general formula of contrastive loss $\mathcal{L}_{contrast}$ shown in Eq. 1 for our derivation. But we note that replacing $\mathcal{L}_{contrast}$ with $\mathcal{L}_{contrast}^{V_1, V_2}$ is straightforward. The overall proof follows a similar derivation introduced in (Oord et al., 2018).

### C.1  SCORE FUNCTION AS DENSITY RATIO ESTIMATOR

We first show that the optimal score function $h_\theta^*(\{v_1, v_2\})$ that minimizes Eq. 1 is proportional to the density ratio between joint distribution and product of marginals, shown as Eq. 5. For notation convenience, we denote $p(v_1, v_2)$ as data distribution $p_d(\cdot)$ and $p(v_1)p(v_2)$ as noise distribution $p_n(\cdot)$. The loss in Eq. 1 is indeed a cross-entropy loss of classifying the correct positive pair out from the given set $S$. Without loss of generality, we assume the first pair $(v_1^0, v_2^0)$ in $S$ is positive or congruent and all others $(v_1^i, v_2^i), i = 1, 2, ..., k$ are negative or incongruent. The optimal probability for the loss, $p(pos = 0|S)$, should depict the fact that $(v_1^0, v_2^0)$ comes from the data distribution $p_d(\cdot)$ while all other pairs come from the noise distribution $p_n(\cdot)$. Therefore,

$$
\begin{aligned}
p(pos = 0|S) &= \frac{p_d(v_1^0, v_2^0) \prod_{i=1}^k p_n(v_1^i, v_2^i)}{\sum_{j=0}^k p_d(v_1^j, v_2^j) \prod_{i \neq j} p_n(v_1^i, v_2^i)} \\
&= \frac{p(v_1^0, v_2^0) \prod_{i=1}^k p(v_1^i)p(v_2^i)}{\sum_{j=0}^k p(v_1^j, v_2^j) \prod_{i \neq j} p(v_1^i)p(v_2^i)} \\
&= \frac{\frac{p(v_1^0, v_2^0)}{p(v_1^0)p(v_2^0)}}{\sum_{j=0}^k \frac{p(v_1^k, v_2^k)}{p(v_1^k)p(v_2^k)}}
\end{aligned}
$$

where we plug in the definition of $p_d(\cdot)$ and $p_n(\cdot)$, and divide $\prod_{i=0}^k p(v_1^i)p(v_2^2)$ for both the numerator and denominator. By comparing above equation with the loss function in Eq. 1, we can see that the optimal score function $h_\theta^*(\{v_1, v_2\})$ is proportional to the density ratio $\frac{p(v_1, v_2)}{p(v_1)p(v_2)}$. The above derivation is agnostic to which layer the score function starts from, e.g., $h$ can be defined on either the raw input $(v_1, v_2)$ or the latent representation $(z_1, z_2)$. As we care more about the property of the latent representation, for the following derivation we will use $h_{W_{12}}^*(\{z_1, z_2\})$, which is proportional to $\frac{p(z_1, z_2)}{p(z_1)p(z_2)}$.

## C.2 MAXIMIZING LOWER BOUND ON MI

Now we substitute the score function in Eq. 1 with the above density ratio, and the optimal loss objective $\mathcal{L}_{contrast}^{opt}$ becomes:

$$
\begin{aligned}
\mathcal{L}_{contrast}^{opt} &= -\underset{S}{\mathbb{E}} \log \left[ \frac{h_{W_{12}}^*(\{z_1^0, z_2^0\})}{\sum_{i=0}^k h_{W_{12}}^*(\{z_1^i, z_2^i\})} \right] \\
&= -\underset{S}{\mathbb{E}} \log \left[ \frac{\frac{p(z_1^0, z_2^0)}{p(z_1^0)p(z_2^0)}}{\sum_{i=0}^k \frac{p(z_1^i, z_2^i)}{p(z_1^i)p(z_2^i)}} \right] \\
&= \underset{S}{\mathbb{E}} \log \left[ 1 + \frac{p(z_1^0)p(z_2^0)}{p(z_1^0, z_2^0)} \sum_{i=1}^k \frac{p(z_1^i, z_2^i)}{p(z_1^i)p(z_2^i)} \right] \\
&\approx \underset{S}{\mathbb{E}} \log \left[ 1 + \frac{p(z_1^0)p(z_2^0)}{p(z_1^0, z_2^0)} k \underset{z_1}{\mathbb{E}} \left[ \frac{p(z_1|z_2)}{p(z_1)} \right] \right] \\
&= \underset{S}{\mathbb{E}} \log \left[ 1 + \frac{p(z_1^0)p(z_2^0)}{p(z_1^0, z_2^0)} k \right] \\
&\geq \log(k) - \underset{S}{\mathbb{E}} \log \left[ \frac{p(z_1^0, z_2^0)}{p(z_1^0)p(z_2^0)} \right] \\
&= \log(k) - \underset{(z_1, z_2) \sim p_{z_1, z_2}(\cdot)}{\mathbb{E}} \log \left[ \frac{p(z_1, z_2)}{p(z_1)p(z_2)} \right] \\
&= \log(k) - I(z_1; z_2)
\end{aligned}
$$

Therefore, for any two views $V_i$ and $V_j$, we have $I(z_i; z_j) \geq \log(k) - \mathcal{L}_{contrast}^{opt}(V_i, V_j)$. As the $k$ increases, the approximation step becomes more accurate. Given any $k$, minimizing $\mathcal{L}_k(V_i, V_j)$ maximizes the lower bound on the mutual information $I(z_i; z_j)$. We should note that increasing $k$ to infinity does not always lead to a higher lower bound. While $\log(k)$ increases with a larger $k$, the optimization problem becomes harder and $\mathcal{L}_k(V_i, V_j)$ also increases.

## D  IMPLEMENTATION DETAILS

### D.1  STL-10

For a fair comparison with DIM (Hjelm et al., 2019) and CPC (Oord et al., 2018), we adopt the same architecture as that used in DIM and split it into two encoders, each shown as in Table 8. For the implementation of the score function, we adopt similar "encoder-and-dot-product" strategy, which is tantamount to a bilinear model.

In the patch-based contrastive learning stage, we use Adam optimizer with an initial learning rate of $0.001$, $\beta_1 = 0.5$, $\beta_2 = 0.999$. We train for a total of 200 epochs with learning rate decayed by $0.2$ after 120 and 160 epochs. In the non-linear classifier evaluation stage, we use the same optimizer setting. For the NCE-based contrastive learning stage, we train for 320 epochs with the learning rate initialized as $0.03$ and further decayed by 10 for every 40 epochs after the first 200 epochs. The temperature $\tau$ is set as $0.1$. In general, $\tau \in [0.05, 0.2]$ works reasonably well.

### D.2  IMAGENET

For patch-based contrastive loss, we use the same optimizer setting as in Sec. D.1 except that the learning rate is initialized as $0.01$.

For NCE-basd contrastive loss in both full ImageNet and ImageNet100 experiments present in Sec. A.1.2, the encoder architecture used for either L or ab channels is shown in Table 9. In the unsupervised learning stage of AlexNet, we use SGD to train the network for a total of 400 epochs. The temperature $\tau$ is set as $0.07$ by following previous work (Wu et al., 2018). The learning rate is initialized as $0.03$ with a decay of 10 for every 50 epochs after the first 250 epochs. Weight decay is

| Half of AlexNet(Krizhevsky et al., 2012) for STL-10 | | | | | |
|---|---|---|---|---|---|
| Layer | X | C | K | S | P |
| data | 64 | * | – | – | – |
| conv1 | 64 | 48 | 3 | 1 | 1 |
| pool1 | 31 | 48 | 3 | 2 | 0 |
| conv2 | 31 | 96 | 3 | 1 | 1 |
| pool2 | 15 | 96 | 3 | 2 | 0 |
| conv3 | 15 | 192 | 3 | 1 | 1 |
| conv4 | 15 | 192 | 3 | 1 | 1 |
| conv5 | 15 | 96 | 3 | 1 | 1 |
| pool5 | 7 | 96 | 3 | 2 | 0 |
| fc6 | 1 | 2048 | 7 | 1 | 0 |
| fc7 | 1 | 2048 | 1 | 1 | 0 |
| fc8 | 1 | 64 | 1 | 1 | 0 |

Table 8: **The variant of AlexNet architecture used in our CMC for STL-10 (only half is present here due to splitting).** **X** spatial resolution of layer, **C** number of channels in layer; **K** `conv` or `pool` kernel size; **S** computation stride; **P** padding; * channel size is dependent on the input source, e.g. 1 for L channel and 2 for ab channel.

set as $10^{-4}$ and momentum is kept as $0.9$. For the linear classification stage, we train for 160 epochs. The learning rate is initialized as $0.1$ and decayed by $0.2$ every 20 epochs after the first 100 epochs. We set weight decay as $0$ and momentum as $0.9$.

For ResNets in CMC stage, there are three differences. First, we use larger learning rate, that is, we set a base learning rate of 0.03 for every 128 images and then roughly scale it up with the batch size. Specifically, we train: (1) ResNet-50 with $bsz = 280$ and $lr = 0.08$; (2) ResNet-101 with $bsz = 200$ and $lr = 0.05$; (3) ResNet-50 x2 with $bsz = 156$ and $lr = 0.04$. Second, we only train for 280 epochs with learning rate decayed at 160, 200, and 240 epochs. Third, we used Fast Autoaugment (Lim et al., 2019) as data augmentation. In the linear evaluation stage, we train for 100 epochs. The learning rate is initialized as 30 for ResNet-50 and ResNet-101, and 50 for ResNet-50 x2. It is decayed by $0.2$ every 15 epochs after the first 60 epochs. We set weight decay as $0$ and momentum as $0.9$.

| Half of AlexNet(Krizhevsky et al., 2012) for ImageNet | | | | | |
|---|---|---|---|---|---|
| Layer | X | C | K | S | P |
| data | 224 | * | – | – | – |
| conv1 | 55 | 48 | 11 | 4 | 2 |
| pool1 | 27 | 48 | 3 | 2 | 0 |
| conv2 | 27 | 128 | 5 | 1 | 2 |
| pool2 | 13 | 128 | 3 | 2 | 0 |
| conv3 | 13 | 192 | 3 | 1 | 1 |
| conv4 | 13 | 192 | 3 | 1 | 1 |
| conv5 | 13 | 128 | 3 | 1 | 1 |
| pool5 | 6 | 128 | 3 | 2 | 0 |
| fc6 | 1 | 2048 | 6 | 1 | 0 |
| fc7 | 1 | 2048 | 1 | 1 | 0 |
| fc8 | 1 | 128 | 1 | 1 | 0 |

Table 9: **AlexNet architecture used in CMC for ImageNet (only half is present here due to splitting).** **X** spatial resolution of layer, **C** number of channels in layer; **K** `conv` or `pool` kernel size; **S** computation stride; **P** padding; * channel size is dependent on the input source, e.g. 1 for L channel and 2 for ab channel.

### D.3  UCF101 AND HMDB51

Following previous work (Misra et al., 2016; Lee et al., 2017; Sayed et al., 2018; Buchler et al., 2018), we use CaffeNet for the video experiments. We tailor the network and use features from the fc6 layer for contrastive learning. Dropout of $0.5$ is used to alleviate overfitting.

### D.4  NYU DEPTH-V2

While experimenting with different views on NYU Depth-V2 dataset, we encode the features from patches with a size of $128 \times 128$. The detailed architecture is shown in Table 10. In the unsupervised training stage, we use Adam optimizer with an initial learning rate of 0.001, $\beta_1 = 0.5$, $\beta_2 = 0.999$. We train for a total of 3000 epochs with learning rate decayed by 0.2 after 2000, 2400, and 2800 epochs. For the downstream semantic segmentation task, we use the same optimizer setting but train for fewer epochs. We only train 200 epochs for CMC pre-trained models, and train 1000 epochs for the *Random* and *Supervised* baselines until convergence. For the classification task evaluated on STL-10, we use the same optimizer setting as in Sec. D.1 to report numbers in Table 2.

| Encoder Architecture on NYU | | | | | |
|:---:|:---:|:---:|:---:|:---:|:---:|
| **Layer** | **X** | **C** | **K** | **S** | **P** |
| **data** | 128 | * | – | – | – |
| **conv1** | 64 | 64 | 8 | 2 | 3 |
| **pool1** | 32 | 64 | 2 | 2 | 0 |
| **conv2** | 16 | 128 | 4 | 2 | 1 |
| **conv3** | 8 | 256 | 4 | 2 | 1 |
| **conv4** | 8 | 256 | 3 | 1 | 1 |
| **conv5** | 4 | 512 | 4 | 2 | 1 |
| **fc6** | 1 | 512 | 4 | 1 | 0 |
| **fc7** | 1 | 256 | 1 | 1 | 0 |

Table 10: **Encoder architecture used in our CMC for playing with different views on NYU Depth-V2**. **X** spatial resolution of layer, **C** number of channels in layer; **K** `conv` or `pool` kernel size; **S** computation stride; **P** padding; * channel size is dependent on the input source, e.g. 1 for L, 2 for ab, 1 for depth, 3 for surface normal, and 1 for segmentation map.

