# OpenReview forum: "Contrastive Multiview Coding"
_ICLR.cc/2020/Conference — Reject_

### Official Review · AnonReviewer3 · 2019-10-21
**Official Blind Review #3**

**Rating:** 3

**Review:**

The paper presented a multi-view learning method that is based on negative sampling in contrastive learning. The core idea is to set an anchor view and the sample positive and negative data points from the other view and maximise the agreement between positive pairs in learning from two views. When more than two views are presented, the learning objective is a sum over all possible combinations of two views. The performance of the proposed model is good, and the ablation study is interesting.

Comments:

1. The core concept, or at least one of the core concepts, in multi-view learning is the conditional independence.

Normally, the underlying assumption in multi-view learning is that, given the class label, the samples from multiple views are conditionally independent from each other. Therefore, the goal is to learn distinctive representations from different data sources/disjoint populations, so then after learning, the ensemble of them is able to capture a set of diverse aspects of the data. A "side-effect" of learning from multiple views is that individual views indeed get improved by learning from others. Meanwhile, self-supervised learning is the case when the input data to the designed learning system is also the target of the system.

The paper presented an idea for self-supervised learning from multiple views, which is not exactly the same, but still in the same regime. This concept could be used to explain some empirical findings in this paper. Since it is expected, there is even no need in conducting experiments.




2. My main concern of this paper is the novelty, however, the empirical results are strong.

The paper mainly presented a simple yet effective method for self-supervised learning from two views, and the generalisation is a sum over all possible combinations of two views. The method itself has already been proposed many years ago as mentioned in the related work section in the paper, and the generalisation was also described in prior work, which makes me doubt the novelty of the paper.

The earliest work to the best of my knowledge is [1], and later on there are a couple workshops [2,3] on multi-view learning which largely settled the field of learning from multiple views from neural networks', kernels', and bayesian perspectives. Many things mentioned in this paper have already been discovered at that time.

3. The theoretical justification is not as strong as the generalised CCA.

CCA has been applied in the field of multi-view learning and self-supervised learning for long, and it was initially proposed for comparing the correlation between two sets of samples of two random variables. A successful generalisation is the generalised CCA which is capable of learning from multiple views. The formula of GCCA as referred in [4] is simple and elegant, and then the extension of using neural networks is also straightforward. Since people has relatively clearer understanding of CCA itself, the generalised version or the kernel version of it is also well-understood.

A nice theoretical understanding of contrastive unsupervised learning is provided in [5], and I recommend the authors to study.


[1] de Sa, Virginia R. "Learning classification with unlabeled data." Advances in neural information processing systems. 1994.
[2] ICML Workshop, "Learning With Multiple Views". 2005
[3] NIPS workshop, "Learning from multiple sources". 2008
[4] Benton, Adrian, et al. "Deep generalized canonical correlation analysis." ICLR workshop 2017.
[5] Arora, Sanjeev, et al. "A theoretical analysis of contrastive unsupervised representation learning." ICML 2019.

**Experience Assessment:**

I have published one or two papers in this area.

**Review Assessment: Checking Correctness Of Derivations And Theory:**

I carefully checked the derivations and theory.

**Review Assessment: Checking Correctness Of Experiments:**

I carefully checked the experiments.

**Review Assessment: Thoroughness In Paper Reading:**

I read the paper thoroughly.

---

> ### Author Response · Authors · 2019-11-08
> **Response to Reviewer 3**
>
> Dear Reviewer 3,
>
> Thank you for your constructive review.
>
> We agree that many methods for multiview learning have been developed since the 1990s. Here we are not claiming the first framework or theory for unsupervised multiview learning, rather we want to empirically illustrate that multiview learning methods (instantiated as contrastive learning here) can beat recent state of the art self-supervised methods, specifically in a large- scale setting, e.g., ImageNet. Our paper further contributes experiments that explicate various properties of multiview learning in the large-scale setting, such as the relative performance of contrastive versus predictive objectives, and the relationship between mutual information between views and the quality of representations learned from these views.
>
> 1. We agree that the concept of conditional independence might explain some of the empirical results. We want to clarify that during our unsupervised training stage, we did not condition on labels, which self-supervised methods assume not to be available.
>
> “This concept could be used to explain some empirical findings in this paper. Since it is expected, there is even no need in conducting experiments”.
> The connection is not that clear to us at this point. Would you please point to a reference such that we can see the connection?
>
> “Meanwhile, self-supervised learning is the case when the input data to the designed learning system is also the target of the system.”
> We want to clarify that our target is not to predict the input (predictive way), rather it is instantiated in a contrastive way, which yields significantly better results than the predictive approach, as shown in the paper.
>
> 2. Thank you very much for pointing us to [1], which directly relates to our work and we are more than happy to add a citation to it (note we did point to another of De Sa’s other papers which also shares similar ideas). We agree that the high-level idea of leveraging co-occurrence is similar, but the learning objectives and detailed instantiation are very different. The update rule of [1], as shown in Figure 6 of [1], is different from our current SGD-based update rule, and it seems difficult to implement in modern deep networks with large-scale data. Indeed, different learning objectives can make a big difference in performance. For example, another previous work [a] did cross-view prediction, while we do cross-view contrastive learning. Our objective leads to a significant improvement over cross-view prediction, (e.g., our objective achieves 42.6% accuracy on ImageNet and 86.88% accuracy on STL-10, while cross-view prediction gives 35.4% and 72.35% accuracies, respectively).
>
> [a] Split-brain autoencoders: Unsupervised learning by cross-channel prediction. CVPR 2017
>
> “The method itself has already been proposed many years ago as mentioned in the related work section in the paper, and the generalisation was also described in prior work.”
> Would you please point to us which older methods are identical to or almost the same as ours? As discussed in our paper, our method is indeed an extension of 2018’s Contrastive Predictive Coding, to the multiview setting, but we are not aware of earlier work that uses the same specific formulation (we also looked at the workshops [2] and [3] pointed out by you).
>
> 3. We agree that CCA has a solid theoretical justification, but this does not imply mutual information maximization is not well justified. Our conjecture is that, capturing mutual information between the latent representations of two views brings about more powerful representations than only capturing their linear correlations as CCA does. Thank you for pointing out DGCCA, which indeed we overlooked and will cite in the revision. We performed an experiment testing the transfer performance of DGCCA on STL-10. We find that DGCCA only yields 22.8% accuracy, which is significantly lower than the 86% accuracy achieved by our method. One reason for the poor performance of DGCCA on this experiment may be that we were only able to use a small batch size (128 images) due to memory constraints. In the original DGCCA paper, a larger batch size (>= 2000) was used but only demonstrated on text datasets, which are less memory intensive than image datasets.  We feel it would be an interesting direction to adapt DGCCA to be effective on large-scale image datasets, but consider this to be non-trivial and out of the scope for our current paper.

---

### Official Review · AnonReviewer2 · 2019-10-23
**Official Blind Review #2**

**Rating:** 6

**Review:**

This paper proposed a new self-supervised learning methods by utilizing contrastive predictive coding technique.  The proposed algorithm is more effective than existing self-supervised learning algorithm.  The presented results are encouraging.
1. In section 3.2,  the authors show that  a large number of views would improve the representation quality. However,  multi-views may provide redundancy information. What is the core information that affect  the representation quality?

In fact,  I am not an expert on self-supervised learning and  contrastive predictive coding,  so my reviewer confidence is low.

**Experience Assessment:**

I do not know much about this area.

**Review Assessment: Checking Correctness Of Derivations And Theory:**

I did not assess the derivations or theory.

**Review Assessment: Checking Correctness Of Experiments:**

I assessed the sensibility of the experiments.

**Review Assessment: Thoroughness In Paper Reading:**

I made a quick assessment of this paper.

---

> ### Author Response · Authors · 2019-11-12
> **Response to Reviewer 2**
>
>
> Dear Reviewer 2,
>
> Thank you very much for your review. We would like to explain more about our intuition here.
>
> “However,  multi-views may provide redundancy information. What is the core information that affect  the representation quality?”
>
> Our hypothesis is that each view has two parts of information: (a) nuisance factors, like sensor noise, that can not be predictive of other views, and (b) information shared with other views. Our learning objective (see Eq.2 and Eq.6) asks the learned latent representation to focus on part (b) such that mutual information between views gets maximized.
>
> Moreover, for each view, the information bits in part (b) are not equal. Some information bits, such as the information of object category (e.g., dog), are shared by many views, while some are shared by only a few. Therefore, if we contrast one single view with many other views, each bit of part (b) will be ordered by the number of times it is shared with those contrasted views. Our conjecture is that the category-level semantics tend to be shared across many views, and thus are prioritized by our method. As a result, the learned representations convey sufficient semantic information.
>
> Therefore, we are leveraging the redundant information between different views/modalities to educate  or teach each other. This mechanism actually has been explored in the field of developmental psychology. One such reference is [a], which argues that human infants utilize the redundancy between the senses in order to build up representations that are mutually predictive of each other. Indeed, if there is no redundant information across views, we cannot learn a good representation in such a way.
>
> [a] Linda Smith. The Development of Embodied Cognition: Six Lessons from Babies. 2005.
>
> Please don’t hesitate to let us know for any further feedback. Thanks!

---

### Official Review · AnonReviewer1 · 2019-10-23
**Official Blind Review #1**

**Rating:** 6

**Review:**

This interesting paper on an important topic; however, its readability could be dramatically improved, especially for the reader less familiar with the problem.

In order to make the paper more accessible, the authors should reorganize the introduction by breaking it down into two parts:
1) a more traditional introduction
- one intuitive paragraph about multi-view coding
- one intuitive paragraph with an illustrative example on how the proposed approach will help solve a problem; at the same intuitive level, compare-and-contrast it with existing approaches
- one intuitive, in-detail paragraph on how the proposed approach works
- one paragraph summarizing the main findings/results
2) a second, new section, that will turn the current Figures 1 & 2 into a complete description of an illustrative example (the current, detailed "captions" are a good start, but they should be fleshed out into a full, detailed section of the paper)


**Experience Assessment:**

I have read many papers in this area.

**Review Assessment: Checking Correctness Of Derivations And Theory:**

I did not assess the derivations or theory.

**Review Assessment: Checking Correctness Of Experiments:**

I assessed the sensibility of the experiments.

**Review Assessment: Thoroughness In Paper Reading:**

I read the paper at least twice and used my best judgement in assessing the paper.

---

> ### Author Response · Authors · 2019-11-12
> **Response to Reviewer 1**
>
> Dear Reviewer 1,
>
> Thank you for the constructive suggestions.
>
> We will take your advice into account as we revise the paper, and in particular are working to make the introduction clearer, and to state up front concretely  what we do. We will upload a revised version once it’s available.
>
> Please don’t hesitate to let us know for any additional comments. Thank you!

---

### Decision · Program_Chairs · 2019-12-19

**Decision:**

Reject

**Comment:**

This paper proposes to use contrastive predictive coding for self-supervised learning.  The proposed approach is shown empirically to be  more effective than existing self-supervised learning algorithms.  While the reviewers found the experimental results encouraging, there were some questions about the contribution as a whole, in particular the lack of theoretical justification.